

# Mechanisms of diversity maintenance in dung beetle assemblages in a heterogeneous tropical landscape

Jose D. Rivera[1,2], Benigno Gómez[1], Darío A. Navarrete-Gutiérrez[3],
Lorena Ruíz-Montoya[1], Leonardo Delgado[4] and Mario E. Favila[2]

[1] Departamento Conservación de la Biodiversidad, El Colegio de la Frontera Sur, San Cristóbal de las Casas, Chiapas, Mexico
[2] Red de Ecoetología, Instituto de Ecología, A.C., Xalapa-Enriquez, Veracruz, Mexico
[3] Departamento: Observación y Estudio de la tierra, Atmósferay Oceano (TAO). Grupo academico: Ecología, paisaje y sustentabilidad, El Colegio de la Frontera Sur, San Cristóbal de las Casas, Chiapas, Mexico
[4] Red de Biodiversidad y Sistemática, Instituto de Ecología, A.C., Xalapa-Enriquez, Veracruz, Mexico

Corresponding author
Jose D. Rivera, jdr495@hotmail.com

## ABSTRACT

**Background**. Anthropized landscapes play a crucial role in biodiversity conservation, as they encompass about 90% of the remaining tropical forest. Effective conservation strategies require a deep understanding of how anthropic disturbances determine diversity patterns across these landscapes. Here, we evaluated how attributes and assembly mechanisms of dung beetle communities vary across the Selva El Ocote Biosphere Reserve (REBISO) landscape.

**Methods**. Community attributes (species diversity, abundance, and biomass) were assessed at the landscape scale, using spatial windows and vegetation classes. Windows were categorized as intact, variegated, or fragmented based on their percent cover of tropical forest. The vegetation classes analyzed were tropical forest, second-growth forest, and pastures.

**Results**. We collected 15,457 individuals and 55 species. Variegated windows, tropical forests, and second-growth forests showed the highest diversity values, while the lowest values were found in intact windows and pastures. Landscape fragmentation was positively and strongly related to dung beetle diversity and negatively related to their abundance; biomass was positively associated with forest cover. Beta diversity was the primary driver of the high dung beetle diversity in the landscape analyzed.

**Discussion**. The landscape heterogeneity and its biodiversity-friendly matrix facilitate the complementarity of dung beetle assemblages in the Selva El Ocote Biosphere Reserve. Random processes govern beta diversity patterns in intact and variegated windows. Therefore, vegetation cover in the region is sufficient to maintain a continuous flow of dung beetles between forested landscape segments. However, intense anthropic disturbances acted as deterministic environmental filters in fragmented windows and pastures sites, leading to biotic homogenization processes. Our results suggest that increasing habitat variegation in highly fragmented sites is an effective strategy to prevent or buffer homogenization processes in the REBISO landscape.

## INTRODUCTION

Anthropized neotropical landscapes encompass a complex combination of natural and semi-natural habitats, where some species can thrive while others may go locally extinct (*De Castro Solar et al., 2015*). Today, almost 90% of remaining tropical forests are located within anthropized landscapes (*Chazdon et al., 2009*). These landscapes now play a crucial role in biodiversity conservation agendas (*De Clerck et al., 2010*). Therefore, it is imperative to understand how species diversity responds to anthropized landscapes in order to implement suitable management actions (*Gardner et al., 2009*; *Socolar et al., 2016*), especially given the multiple successional pathways and disturbance states that these modified landscapes can follow (*Fischer & Lindenmayer, 2007*; *Arroyo-Rodríguez et al., 2017*).

Traditionally, researchers have assessed the effect of anthropic disturbance on biotic communities by comparing one or more community attributes (e.g., species diversity, abundance, biomass) across different sampling units at a local level (i.e., vegetation cover types or land-use types). However, the composition and configuration of the habitats that surround the sampling units are also important drivers of ecological processes in biotic communities (*Franklin & Lindenmayer, 2009*). A landscape-level approach provides the necessary context to understand better how communities respond to anthropic disturbances by incorporating the effects of the multiple landscape components (*Gardner et al., 2009*; *Hodder et al., 2014*). Besides, landscape studies provide useful information for effective natural resource management since many anthropogenic drivers of biodiversity loss, e.g., land-use change or habitat destruction, operate at the landscape level (*Hodder et al., 2014*). *McIntyre & Barret (1992)* coined the variegation concept for anthropized landscapes exhibiting disturbance and vegetation cover gradients. *McIntyre & Hobbs (1999)* then added the fragmentation concept to the variegation model. These authors classified the landscape into four categories based on the percentage of remaining original vegetation (OV) and the intensity of habitat transformation: (a) intact landscapes (>90% OV): sites with little or no modification; (b) variegated landscapes (60–90% OV), showing either gradual or abrupt limits between their component units; (c) fragmented landscapes (10–60% OV), characterized by a high degree of modification; and d) relict landscapes (<10% OV), showing severe modification and almost no forest cover remnants. *Halffter & Rös (2013)* proposed studying landscape diversity through sampling windows in the geographical space analyzed. These windows are based on the landscape model proposed by *McIntyre & Hobbs (1999)* and consist of equally-sized sampling spaces that are semi-randomly located to maximize the representation of the vegetation heterogeneity and land-use types in the landscape.

The Selva El Ocote Biosphere Reserve (REBISO, hereafter) harbors some of the most heterogeneous, although highly disturbed, remnants of tropical forest in Mexico (*Flamenco-Sandoval, Martínez Ramos & Masera, 2007*). Frequent forest fires, in addition to the complex geological nature, climate features, and socio-economic dynamics (livestock and agricultural activities) in the REBISO have led to a complex landscape comprising a mosaic of tropical forests, second-growth forests, pastures, and croplands (*Ochoa, 1996*; *SEMARNAT/CONANP, 2001*; *Flamenco-Sandoval, Martínez Ramos & Masera, 2007*;

*Ramírez-Marcial et al., 2017*). Thus, a landscape-level approach seems most appropriate for examining how species respond to anthropogenic disturbance in the REBISO, given its complex and heterogeneous landscape.

Dung beetles (Scarabaeidae: Scarabaeinae) are globally distributed insects that feed on decomposing organic matter such as mammal feces, carrion, rotting fruit, or fungi (*Halffter & Matthews, 1966*). Due to their sensitivity to environmental disturbances, dung beetles are ideal bioindicators to assess the effects of landscape changes on diversity (*Favila & Halffter, 1997*; *Nichols et al., 2007*). Previous studies have shown how habitat loss leads to abrupt changes in the composition and structure of dung beetle communities (*Klein, 1989*; *Quintero & Roslin, 2005*; *Nichols et al., 2007*; *Navarrete & Halffter, 2008*; *Díaz, Galante & Favila, 2010*; *Cajaiba et al., 2017*). However, few studies have evaluated the response of dung beetle communities to disturbances at the landscape level (*Numa et al. , 2009*; *Rös, Escobar & Halffter, 2012*; *Sánchez-de Jesús et al., 2016*; *Alvarado et al., 2018*; *Alvarado et al., 2020*), or whether the observed diversity patterns are stochastic or determined by environmental filters or competitive exclusion between species (*Ortega-Martínez et al., 2020*). Assessing dung beetle diversity at the landscape level, using multiple but complementary metrics, can provide a more comprehensive view of how diversity is maintained and what community assembly mechanisms operate in anthropized landscapes.

In this study, we evaluate the assemblage structure and diversity patterns of dung beetle communities in the heterogeneous tropical landscape of the Selva El Ocote Biosphere Reserve. We address the following questions: (1) How do the diversity and structure of dung beetle assemblages vary across the REBISO landscape and its vegetation classes? (2) How do the composition and configuration of the REBISO landscape influence the diversity and structure of dung beetle assemblages? (3) How does beta diversity change and is maintained across the landscape and between different vegetation classes? The information obtained in this study will be useful for designing conservation strategies in complex tropical landscapes with different heterogeneity levels.

## MATERIALS & METHODS

### Study area

The study was carried out at the REBISO, located in the municipalities of Ocozocoautla de Espinosa and Cintalapa, Chiapas, Mexico (16°45′42″–17°09′00″N and 93°54′19″–93°21′20″W, Fig. 1). The area is mostly underlain by dolomite rocks and limestone, with a dominance of water-soluble sedimentary rocks (*Domenici, 2016*). The predominant climate types are warm, humid (climate type Am) and warm, subhumid (climate type Am(f)), with a mean annual temperature of 22 °C and heavy rainfall throughout the year (*SEMARNAT/CONANP, 2001*).

We produced a vegetation map of REBISO from a multispectral SPOT6 image acquired in 2014, using a supervised classification method in QGIS v2.12.3 (*Q GIS Development Team, 2016*). The vegetation classes considered were tropical forest, second-growth forest, and pastures (Fig. 1, Table 1).

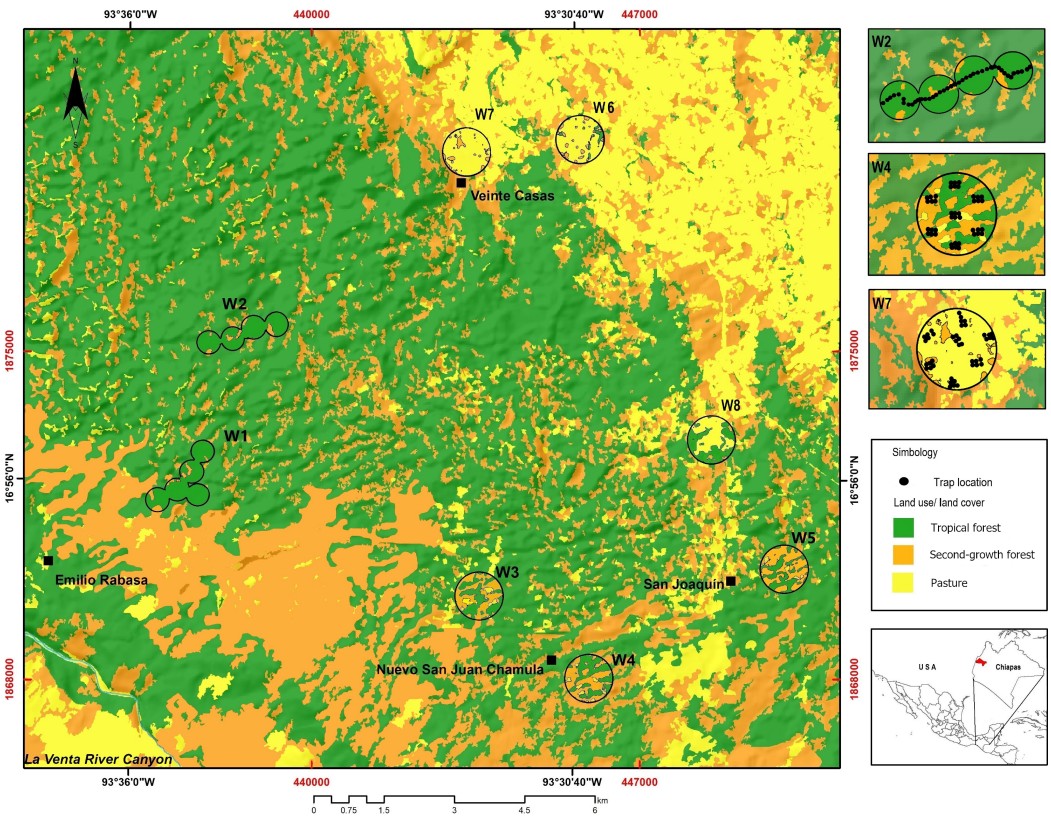

**Figure 1** Study area showing the location of the sampling windows at the Biosphere Reserve Selva El Ocote, Chiapas.

## Sampling design

We established eight 1-km$^2$ (100 ha) sampling windows to capture the landscape heterogeneity in the REBISO (*Sánchez-de Jesús et al., 2016*). Each window was separated from each other by at least 2 km to ensure spatial independence between them (*Sánchez-de Jesús et al., 2016*). The landscape composition in each window was described by estimating the percent coverage of each vegetation class and evaluating the evenness of their distribution (landscape compositional diversity - Shannon diversity). The spatial configuration of the vegetation classes in each window was assessed with the splitting index and edge density metric (*McGarigal, Cushman & Ene, 2012*). Edge density is computed as the length (m) of the edges of each vegetation class divided by the window area (ha). The splitting index describes the degree of fragmentation of a landscape and is equivalent to the effective number of patches. Thus, as a landscape becomes increasingly sub-divided, the splitting index increases (*Jaeger, 2000*; *Fahrig, 2017*). The landscape composition, edge density, and splitting index metrics (Table S1) were obtained with FRAGSTAT v4.2.1 (*McGarigal, Cushman & Ene, 2012*). Based on the percent cover of tropical forest (F), windows were classified as intact (W1, W2; F >90%), variegated (W3, W4, and W5; 60% <F <90%), or fragmented (W6, W7, and W8; 10% <F <60%).

**Table 1** Vegetation classes found at Reserva de la Biosfera Selva El Ocote, Mexico.

| Vegetation class | Description |
| --- | --- |
| Tropical forest ($n = 7$):[a]**38%** **W1, W2, W3, W4, W5, W6, W8.** | Forest in a mature successional stage with a dense canopy cover. The most common tree species are *Pseudolmedia spuria*, *Louteridium donnell-smithii*, *Manilkara sapota*, *Swietenia macrophylla* and *Quararibea funebris* (*SEMARNAT/CONANP, 2001*; *Ramírez-Marcial et al., 2017*). Mean canopy cover, 82.32% ($\pm$1.35 s.e.); mean basal area, 912.24 cm$^2$ ($\pm$163.88 s.e.). |
| Second-growth forest ($n = 8$):[a]**30%** **W1, W2, W3, W4, W5, W6, W7, W8** | Forest in intermediate successional stage, recovering after 1998 fire; the canopy is less dense than in the tropical forest. Dominated by *Heliocarpus appendiculatus* and *Eugenia acapulcensis* (*SEMARNAT/CONANP, 2001*; *Ramírez-Marcial et al., 2017*). Mean canopy cover, 56.76% ($\pm$3.22 s.e.); mean basal area, 577.65 cm$^2$ ($\pm$105.14 s.e.). |
| Pasture ($n = 5$):[a]**32%** **W3, W4, W5, W6, W7, W8** | Pastures are at least ten years old (*SEMARNAT/CONANP, 2001*). The few trees present are used mainly as shade for cattle. Mean basal area, 874.29 cm$^2$ ($\pm$ s.e. 94.60); canopy cover ranges from 2% to 53% ($\bar{x}$ 22.11%, $\pm$s.e 3.03). |

Notes.
[a] Mean coverage over the eight windows.

We sampled dung beetles (Scarabaeidae: Scarabaeinae) during the dry (March to May) and the rainy (July–August) seasons of 2016 using pitfall traps. This sampling scheme allowed us to integrate the seasonal activities of dung beetles (*Cajaiba et al., 2017*). Each trap consisted of a 1 L cylindrical plastic container with 300 mL of ethylene glycol as preservative, buried at ground level, and covered with a plastic lid to protect the bait from rain and sun radiation. Pitfall traps were baited with 70 g of either an 80:20 mixture of pig and human feces (copro-traps) or squid flesh (necro-traps) in order to obtain a representative sample of the dung beetle assemblages in the area.

Seven sampling sites were established in each window, separated 250–360 m from each other, to proportionally adjust the number of pitfall traps per vegetation class according to the vegetation class composition of each window (Table S2). Proportional sampling is a suitable method for detecting changes in beta diversity in heterogeneous landscapes (*Schoereder et al., 2004*). In each sampling site, three copro-traps and three necro-traps (42 traps/window), were placed in a rectangular area separated 50 m from each other to minimize interference between them (*Larsen & Forsyth, 2005*). The rectangular layout of some trap sets was modified in some cases due to the topographic characteristics of the sites. The pitfall traps were left active for 48 h.

The specimens collected were counted and identified to species. To estimate the dung beetle biomass, we randomly selected ten specimens of each species and dried them at 70 °C for 72 h. We weighed each specimen to the nearest 0.1mg with an analytical balance (Explorer Pro) and calculated the average biomass for each species. Finally, we multiplied the mean biomass of each species by its abundance in each window and vegetation class. The dung beetle specimens were deposited in the entomological collection of El Colegio de la Frontera Sur, San Cristóbal de Las Casas. Field sampling in the REBISO was carried out
under permit SGPA/DGS/14214/15 issued by the Secretaria de Medio Ambiente y Recursos Naturales, Mexico.

## Data analysis

We followed a spatial and structural approach (*sensu Rös, Escobar & Halffter, 2012*) to analyze the data. Windows were the sampling units for the spatial approach ($n = 8$), while vegetation classes within windows were the sampling units for the structural approach ($n = 21$). The sampling completeness of each window and vegetation class was determined using the coverage estimator of *Chao & Jost (2012)*, which allows comparing species diversity across multiple sites.

Alpha diversity in each sampling unit (window or vegetation class) was evaluated using the $^0D$ and $^1D$ diversity numbers. $^0D$ is equivalent to species richness and is insensitive to the species abundance (*Jost, 2006*); $^1D$ is equivalent to the exponential of Shannon diversity index and accounts for the most abundant species in a community (*Jost, 2006*).

We examined differences in species richness between windows by constructing and comparing their 95% bootstrap confidence intervals. Non-overlapping confidence intervals denote significantly different species richness (*Gotelli & Colwell, 2011*; *Chao et al., 2014*). Differences in species richness between vegetation classes were determined using interpolation–extrapolation curves (*Chao et al., 2014*). The sampling coverage, $^0D$ and $^1D$ diversity numbers, confidence intervals, and the interpolation–extrapolation curves were obtained with the software iNEXT v2.0.11 (*Hsieh, Ma & Chao, 2016*).

Generalized linear models (GLM) were used to assess differences in abundance and biomass between windows and vegetation classes. The abundance and biomass data approached a normal distribution after logarithmic transformation and were analyzed assuming a Gaussian error distribution (*Crawley, 2013*). Pairwise comparisons using Tukey's test were carried out, with the multcomp package (*Hothorn Bretz & Westfall, 2008*) whenever significant differences were detected.

GLMs were also used to assess the effect of the landscape composition and configuration on the species richness ($^0D$), exponential of the Shannon diversity ($^1D$), abundance, and biomass of the dung beetle assemblages. These data were first tested for normality and were then analyzed assuming a Gaussian error distribution. Since only eight observations were available to fit these models, separate models containing only one predictor variable were constructed to avoid overfitting (*Kelley & Maxwell, 2003*). The best-fit models were selected based on the Akaike's information criterion corrected for small samples (AICc) and the deviance explained ($D^2$). The model with the smallest AICc ($\Delta$AICc >2) and the largest $D^2$ values was selected as the best-fit model (*Burnham & Anderson, 2002*). Based on the results from the Moran I test (as implemented in the package LetsR), no significant spatial structure was detected in the response variables (Table S3) (*Vilela & Villalobos, 2015*).

True beta diversity (i.e., the effective number of distinct communities) was estimated for species richness ($^0\beta$) and Shannon diversity ($^1\beta$) using the multiplicative partitioning method (*Jost, 2007*). The multiple-site *Sørensen* dissimilarity was partitioned as $\beta_{Sor} = \beta_{Sim} + \beta_{Sne}$ using the package Betapart v1.3 (*Baselga & Orme, 2012*) to determine whether

the ecological differences between sampling units resulted from species turnover ($\beta_{Sim}$) or nestedness ($\beta_{Sne}$). Turnover measures the replacement of species between sites caused by environmental differences, disturbance, or competition. Nestedness is a loss of species between sites, usually due to differences in local conditions or ecological niches, where the species-poorer site contains a subset of the species present in the species-richer site (*Legendre, 2014*).

Null models were used to determine whether beta diversity patterns resulted from either random changes in alpha and gamma diversity, or from underlying deterministic mechanisms in communities or the landscape (*Chase et al., 2011*). We constructed null models for the beta Raup-Crick index ($\beta_{R-C}$) using the algorithm developed by *Chase et al. (2011)* with 9999 randomizations. $\beta_{R-C}$ compares the observed versus expected beta diversity under the null model, scaling the results to a range between −1 and 1. This value indicates whether the beta diversity observed between windows, or vegetation classes, is more similar (values close to −1), equal (values close to 0), or less similar (values close to 1) than the one expected by chance ($\beta_{R-C}$ null model). We built a dendrogram and a nonmetric multidimensional scaling (NMDS) plot based on $\beta_{R-C}$ values for windows and vegetation classes, respectively (*Chase et al., 2011*). The dendrogram was constructed using the complete linkage method, as it produces clusters with ecological discontinuities (*Legendre & Legendre, 2003*). We compared the dendrogram and NMDS plot based on $\beta_{R-C}$ with homologous plots based on *Sørensen* dissimilarity to examine whether deterministic mechanisms are underlying the observed beta diversity across the landscape (*Chase et al., 2011*). All statistical analyses and models were carried out using R v.3.3.1 *R Development Core Team (2015)*.

## RESULTS

We collected a total of 15,457 specimens belonging to 55 species in the eight windows at REBISO (Table S4A). The most abundant species was *Deltochilum mexicanum* (15% of total abundance), followed by *Onthophagus corrosus* (13%), *Eurysternus maya* (12%), *Canthon vazquezae* (11%), and *Onthophagus batesi* (8%). Sampling coverage on each window was 99% (Table S4A). However, the sampling coverage of vegetation classes varied between windows: for forest vegetation it ranged from 91% (W6) to 100% (W8), it was over 98% for second-growth forests, and between 95% (W3) and 99% (W6) for pastures (Table S4A).

### Diversity, Abundance, and Biomass Patterns in Windows

Species richness ($^0D$) in the windows sampled ranged from 22 (W1, intact window) to 37 (W4, variegated window), whereas the exponential Shannon diversity index ($^1D$) ranged from 4.9 (W2, intact window) to 17.6 (W5, variegated window) species (Table 2). Species richness in W4 and W5 was significantly higher than in the other windows (Fig. 2A).

*Deltochilum mexicanum*, *E. maya*, and *C. vazquezae* were the most abundant species in intact windows W1 and W2 (Fig. S1A); *C. vazqueze*, *E. maya*, and *Eurysternus angustulus* were the most abundant ones in variegated windows W3, W4, and W5; and *O. batesi*, *O. corrosus*, and *Copris lugubris* were the most abundant species in fragmented windows W6, W7, and W8 (Fig. S1A). Biomass patterns in the dung beetle communities differed

**Table 2** $^0D$ and $^1D$ values in each window and vegetation class at Reserva de la Biosfera Selva El Ocote, Mexico.

| | $^0D$ | | | | $^1D$ | | | |
|---|---|---|---|---|---|---|---|---|
| | F | SF | P | Species richness | F | SF | P | Exp (Shannon diversity) |
| W1 | 22 | 11 | – | 22 | 5.23 | 4.8 | – | 5.36 |
| W2 | 24 | 12 | – | 24 | 5.05 | 4.27 | – | 4.97 |
| W3 | 21 | 21 | 11 | 28 | 7.71 | 7.22 | 5.38 | 8.69 |
| W4 | 31 | 29 | 20 | 37 | 12.39 | 13.51 | 13.91 | 16.49 |
| W5 | 27 | 32 | 19 | 35 | 9.88 | 16.37 | 9.55 | 17.59 |
| W6 | 5 | 14 | 19 | 23 | 3.50 | 5.51 | 6.40 | 6.57 |
| W7 | – | 17 | 19 | 24 | – | 8.08 | 7.38 | 7.61 |
| W8 | 2 | 20 | 21 | 26 | 1.50 | 3.46 | 9.51 | 5.49 |
| $\gamma$ | 45 | 44 | 34 | 55[a]/55[b] | 7.75 | 15.54 | 8.93 | 15.84[a]/15.84[b] |
| $\alpha$ | 18.85 | 19.5 | 18.66 | 27.37[a]/18.9[b] | 2.93 | 6.57 | 7.34 | 7.09[a]/6.15[b] |
| $\beta$ | 2.38 | 2.25 | 1.87 | 2.01[a]/2.9[b] | 2.64 | 2.36 | 1.21 | 2.23[a]/2.57[b] |

**Notes.**

F, Forest; SF, Second-growth forest; P, Pasture.

[a] Overall window diversity.

[b] Overall vegetation class diversity.

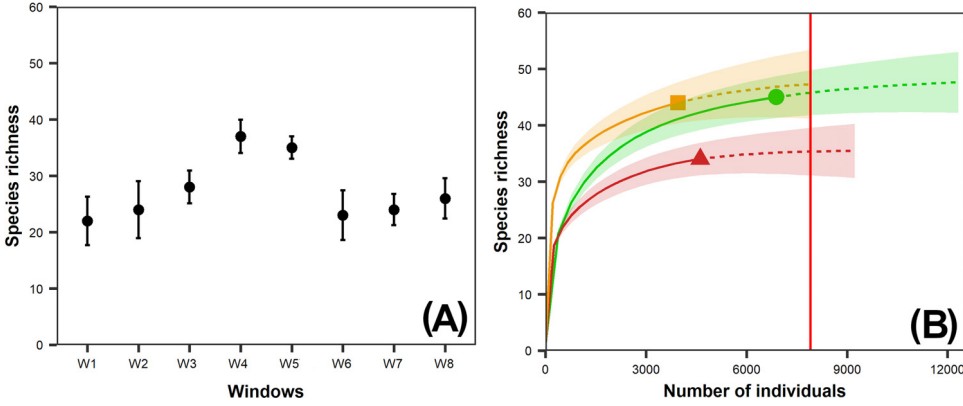

**Figure 2** **(A) Observed species richness of dung beetles per window (W); (B) Interpolation-extrapolation species accumulation curve per vegetation class.** 95% bootstrap confidence intervals are shown for observed richness per window and the species accumulation curves per vegetation class; (A) Intact Landscapes: W1, W2; Variegated Landscapes: W3, W4, W5; Fragmented Landscapes: W6, W7, W8; (B) Circle: Forest; Triangle: Second-growth forest, Square: Pasture. The red vertical line at the end of the species extrapolation curve of the second-growth forest represents the base sample size for comparison of species richness between vegetation classes.

from those observed in their abundance values. *Deltochilum mexicanum*, *E. maya*, and *Ontherus mexicanus* were the dominant species, in terms of biomass, in intact windows W1 and W2 (Fig. S1A); *D. mexicanum*, *E. maya*, and *Dichotomius amplicollis* were the dominant species in W3; *D. mexicanum*, *D. amplicollis*, and *Dichotomius annae* in W4; and *Coprophanaeus corythus*, *Deltochilum sublaeve*, and *D. amplicollis* in W5. *Coprophanaeus*

**Table 3  Mean abundance and biomass (g) per trap (±sd) in each window and vegetation class.**

|  | Mean abundance/trap (±sd) | $P < 0.05$ | Mean biomass/trap (±sd) | $P < 0.05$ |
|---|---|---|---|---|
| **W1** | 52.94 (11.13) | a | 9.07 (1.01) | a |
| **W2** | 44.95 (0.11) | a | 8.20 (0.93) | a, b |
| **W3** | 20.83 (13.94) | a | 2.37 (0.97) | c |
| **W4** | 15.72 (7.88) | a | 2.53 (0.72) | c |
| **W5** | 29.55 (10.88) | a | 3.89 (1.70) | b, c |
| **W6** | 15.82 (5.12) | a | 2.20 (0.49) | c |
| **W7** | 35.26 (18.29) | a | 2.75 (0.05) | c |
| **W8** | 36.51 (46.98) | a | 3.71 (1.22) | b, c |
| **F** | 30.28 (13.91) | a | 5.05 (2.21) | a |
| **SF** | 37.89 (26.86) | a | 4.53 (3.04) | a |
| **P** | 17.74 (15.56) | a | 2.09 (0.62) | b |

Notes.

F, Forest; SF, Second-growth forest; P, Pasture.

[a] Pairwise comparison results are shown in Table S5. Different letters indicate statistically significant differences between windows and between vegetation classes ($P < 0.05$).

*corythus*, *C. lugubris*, and *D. amplicollis* were the species with the highest biomass in fragmented windows W6, W7, and W8 (Fig. S1A).

The highest abundance values (44–52 individuals per trap) were recorded in the intact windows W1 and W2 (Table 3), followed by fragmented windows W6, W7, and W8 (30–36 individuals per trap), and variegated windows W3, W4, and W5 (15–30 individuals per trap). However, these differences were not statistically significant ($\chi^2 = 8.923$; $df = 7$; $P = 0.26$; Table 3). By contrast, there were significant differences in mean biomass between windows ($\chi^2 = 45.143$; $df = 7$; $P => 0.001$; Table 3). Mean biomass per trap was significantly higher in windows W1 and W2 (8.2–9.1 grams per trap, Table 3), but no significant differences were found between fragmented (2.2–3.7 grams per trap, Table 3) and variegated windows (2.4–3.9 grams per trap, Table 3).

All the landscape variables had a significant positive effect on the species richness ($^0D$) in each window. However, the splitting index was the variable that best explained variations in species richness (Table 4). Although the exponential Shannon diversity ($^1D$) values were positively related to the splitting index and edge density, the splitting index was the best predictor for variations in Shannon diversity between windows. Edge density and forest cover were the best predictor variables for dung beetle abundance and biomass, respectively; edge density was negatively correlated with abundance, and forest cover positively correlated with biomass (Table 4).

True beta diversity of orders 0 and 1 indicated two effective communities between windows, $^0\beta$ being slightly smaller than $^1\beta$ (Table 2). The multiple-site Sørensen value calculated for all windows was 0.65 (Fig. 3A "W Total"); 85% of this dissimilarity was due to species turnover ($\beta_{Sim}$) and 15% to nestedness-resultant component ($\beta_{Sne}$). Sørensen dissimilarity for intact windows W1 and W2 was lower than 0.4, mainly due to nestedness ($\beta_{Sne}$) (Fig. 3A). Dissimilarity ranged from 0.3 to 0.45 in variegated windows (W3, W4, W5), and from 0.3 to 0.58 in fragmented windows (W6, W7, W8). In most cases (except

**Table 4  Estimated parameters of the best-fit General Linear Models (GLMs) for the effect of landscape composition and configuration on species richness, diversity, abundance, and biomass of dung beetles.** The best-fit GLMs for each response variable are shown, ordered from the best to the worst. Species richness (0D), exponential Shannon Diversity (1D), abundance (Ab), biomass (Bm).

| GLMs | Parameters | Estimate | S.E. | t value | P-value | AICc | D² |
|---|---|---|---|---|---|---|---|
| $^0D \sim$ SPLIT | Intercept | 18.991 | 0.787 | 24.110 | **<0.001** | 34.955 | 0.95 |
| | SPLIT | 2.696 | 0.215 | 12.520 | **<0.001** | | |
| $^0D \sim$ SHDI | Intercept | 10.693 | 5.44 | 1.965 | 0.097 | 53.513 | 0.47 |
| | SHDI | 8.640 | 2.73 | 3.161 | **0.019** | | |
| $^0D \sim$ ED | Intercept | 18.415 | 3.292 | 5.593 | **0.001** | 54.052 | 0.43 |
| | ED | 0.0761 | 0.025 | 2.99 | **0.024** | | |
| $^0D \sim$ FC | Intercept | 27.842 | 3.221 | 8.643 | **<0.001** | 61.306 | 0.00 |
| | FC | −0.012 | 0.063 | −0.195 | 0.851 | | |
| $^1D \sim$ SPLIT | Intercept | 2.026 | 1.405 | 1.443 | 0.199 | 44.206 | 0.79 |
| | SPLIT | 2.273 | 0.384 | 5.921 | **0.001** | | |
| $^1D \sim$ ED | Intercept | 1.570 | 3.202 | 0.490 | 0.641 | 53.606 | 0.34 |
| | ED | 0.064 | 0.025 | 2.584 | **0.041** | | |
| $^1D \sim$ SHDI | Intercept | −3.447 | 5.951 | −0.579 | 0.583 | 54.947 | 0.22 |
| | SHDI | 6.496 | 2.989 | 2.173 | 0.073 | | |
| $^1D \sim$ FC | Intercept | 9.628 | 2.879 | 3.344 | **0.015** | 59.510 | 0.00 |
| | FC | −0.014 | 0.056 | −0.248 | 0.812 | | |
| Ab $\sim$ ED | Intercept | 4.047 | 0.308 | 13.146 | **<0.001** | 16.136 | 0.28 |
| | ED | −0.006 | 0.002 | −2.387 | 0.05 | | |
| Ab $\sim$ SHDI | Intercept | 4.524 | 0.554 | 8.164 | **<0.001** | 16.968 | 0.20 |
| | SHDI | −0.593 | 0.278 | −2.131 | 0.072 | | |
| Ab$\sim$ FC | Intercept | 3.176 | 0.243 | 13.070 | **<0.001** | 19.952 | 0.00 |
| | FC | 0.005 | 0.005 | 1.123 | 0.305 | | |
| Ab$\sim$ SPLIT | Intercept | 3.648 | 0.314 | 11.629 | **<0.001** | 20.222 | 0.00 |
| | SPLIT | −0.086 | 0.086 | −1.009 | 0.32 | | |
| Bm $\sim$ FC | Intercept | 1.752 | 0.696 | 2.517 | **0.045** | 36.795 | 0.70 |
| | FC | 0.064 | 0.014 | 4.678 | **0.003** | | |
| Bm $\sim$ ED | Intercept | 8.037 | 1.6780 | 4.785 | **0.003** | 43.284 | 0.32 |
| | ED | −0.032 | 0.013 | −2.528 | **0.044** | | |
| Bm$\sim$ SHDI | Intercept | 9.910 | 3.346 | 2.962 | **0.025** | 45.773 | 0.08 |
| | SHDI | −2.970 | 1.681 | −1.767 | 0.127 | | |
| Bm$\sim$ SPLIT | Intercept | 5.114 | 1.851 | 2.763 | **0.032** | 48.624 | 0.00 |
| | SPLIT | −0.302 | 0.506 | −0.597 | 0.572 | | |

**Notes.**
FC, % forest cover; SHDI, landscape diversity; PLAND, percentage of each vegetation class in a window; SPLIT, splitting index; ED, edge density; D2, explained deviance.
Bold styling indicates P-values below 0.05

for W8), the observed Sørensen dissimilarity values were primarily due to species turnover (Fig. 3A).

The dendrogram based on the *Sørensen* distance revealed two main groups (Fig. 3B). The first group includes the fragmented windows (W6, W7, W8), while the second group reveals a gradient of increasing similarity ranging from the intact (W1, W2) to the variegated (W3,

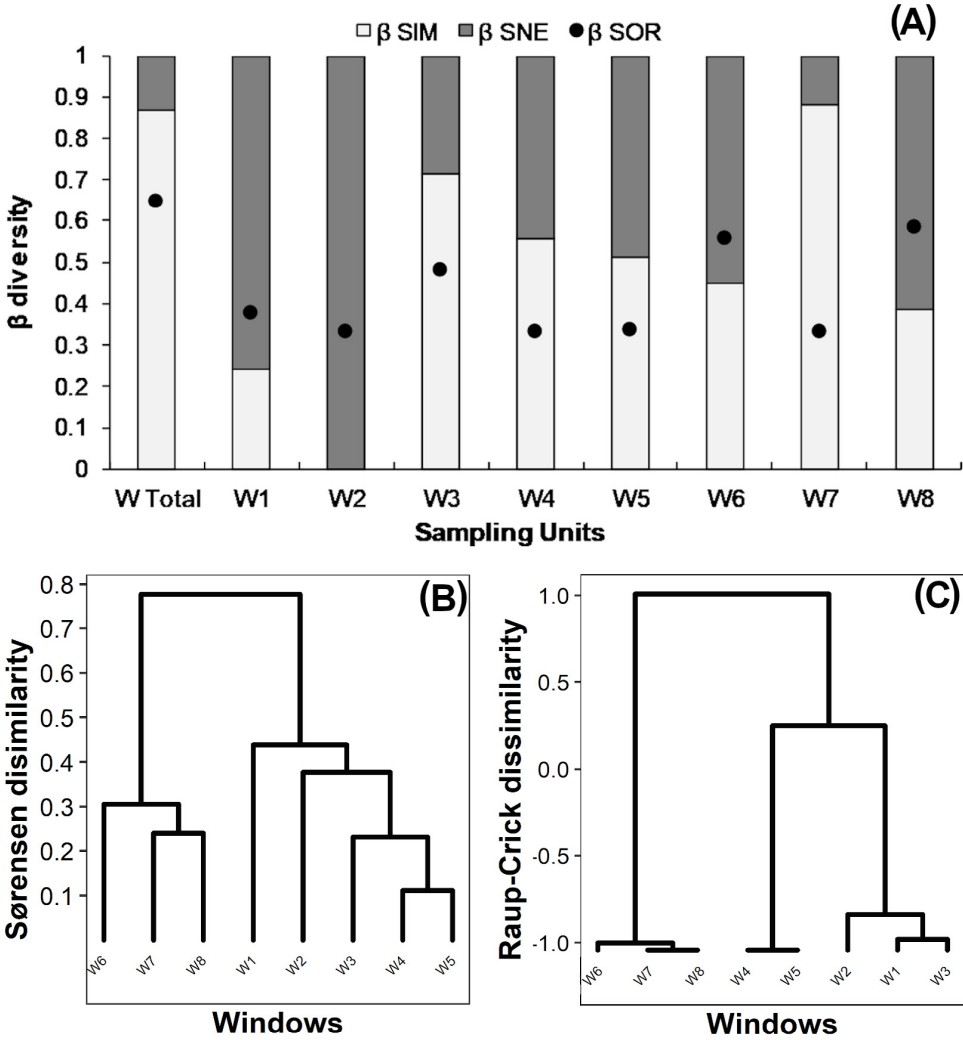

**Figure 3** **(A) Beta diversity within windows; Dendrogram based on (B) Sørensen dissimilarity between windows & (C) βR-C dissimilarity between windows.** (A) Black dots: Sørensen beta diversity (βSor) between windows. White bars: Percentage contribution of species turnover (βSim) to beta diversity (βSim/βSor); Black bars: Percentage contribution of species-nestedness (βSne) to beta diversity (βSne/βSor).

W4, W5) windows. The null-model analysis showed that the difference between fragmented windows with respect to the variegated and intact windows was higher than expected by chance ($\beta_{RC}$ Value: 1.0, Fig. 3C). However, the dissimilarity between variegated and intact windows did not exceed the null expectation of beta diversity ($0 < \beta_{RC} < 0.3$, Fig. 3C).

## Diversity, abundance, and biomass in vegetation classes

The dung beetle species richness ($^0D = 34$ species) in pastures was significantly lower than in the other vegetation classes, but there were no significant differences between second-growth and tropical forests (44 and 45 species, respectively) (Fig. 2B). We recorded

the lowest exponential Shannon diversity values ($^1D$) in the tropical forests (7.75) and the highest in second-growth forests (15.54) (Table 2).

*Deltochilum mexicanum, C. vazquezae,* and *E. maya* were the most abundant species in tropical forest sites (Fig. S1B); *O. corrosus, D. mexicanum,* and *E. maya* in the second-growth forest; and *O. batesi, O. corrosus,* and *C. lugubris* in pasture sites (Fig. S1B). *Deltochilum mexicanum, E. maya,* and *O. mexicanus* contributed with the highest biomass in tropical forests; *D. mexicanum, C. corythus,* and *E. maya* in the second-growth forests (Fig. S1B); and *C. corythus, C. lugubris,* and *D. amplicollis* in pasture sites (Fig. S1B).

No significant differences in abundance were observed between vegetation classes ($\chi^2 = 3.701$; $df = 2$; $P = 0.16$, Table 3). The average number of individuals per trap was 37.8 in second-growth forests, followed by tropical forests (30.2) and pasture sites (17.7) (Table 3). By contrast, there were significant differences in mean biomass between vegetation classes ($\chi^2 = 10.829$; $df = 2$; $P = 0.004$). Pasture sites had a significantly lower mean biomass per trap (2.09 g) than tropical forest (5.05 g) and second-growth forest (4.53) sites, which showed no significant differences between them (Table 3).

According to the multiplicative partition of diversity, there were 2.9 effective communities for $^0\beta$ and 2.5 communities for $^1\beta$ in the three vegetation classes combined. Two effective communities were estimated for both the tropical forest and second-growth forest classes, with $^1\beta$ higher than $^0\beta$ in both cases. Only one effective community was estimated for the pasture class, with $^0\beta$ higher than $^1\beta$ (Table 2). The Sørensen dissimilarity between vegetation classes was 0.85, with 88% of this value accounted for by species turnover ($\beta_{Sim}$) and 12% by nestedness processes ($\beta_{Sne}$) (Fig. 4B).

Tropical forests and second-growth forests showed higher Sørensen values (0.71 and 0.70, respectively) than pastures (0.54) (Fig. 4A). The NMDS plot based on the Sørensen distance formed a compact cluster of pasture sites, whereas most of the tropical forest and second-growth forest sites overlapped between themselves and with the pasture sites (Fig. 4B). The NMDS plot based on the beta Raup-Crick null model index ($\beta_{R-C}$) separated the tropical forest sites from pastures, whereas second-growth forest sites overlapped with tropical forest and pasture classes (Fig. 4C).

## DISCUSSION

Our results identify the REBISO as one of the regions with highest diversity of Scarabaeinae in Mexican tropical forests, with 55 species, along with the Chimalapas, Oaxaca, with 74 species (*Peralta Moctezuma, 2019*); the Lacandon forest, Chiapas, with 49 species (*Navarrete & Halffter, 2008*); and the Tuxtlas forest, Veracruz, with 44 species (*Favila, 2005*).

### Local patterns of species richness and assemblage structure

Dung beetle communities in variegated windows showed the highest richness values in the REBISO. *Rös, Escobar & Halffter (2012)* and *Costa et al. (2017)* also found a higher richness of dung beetle species in variegated landscapes. Landscape variegation can be a significant environmental driver of local diversity as it increases the range of habitats available for species by creating a complex composition and configuration (*Tscharntke et al., 2012*; *Ramírez-Ponce et al., 2019*). The high species richness and diversity found in

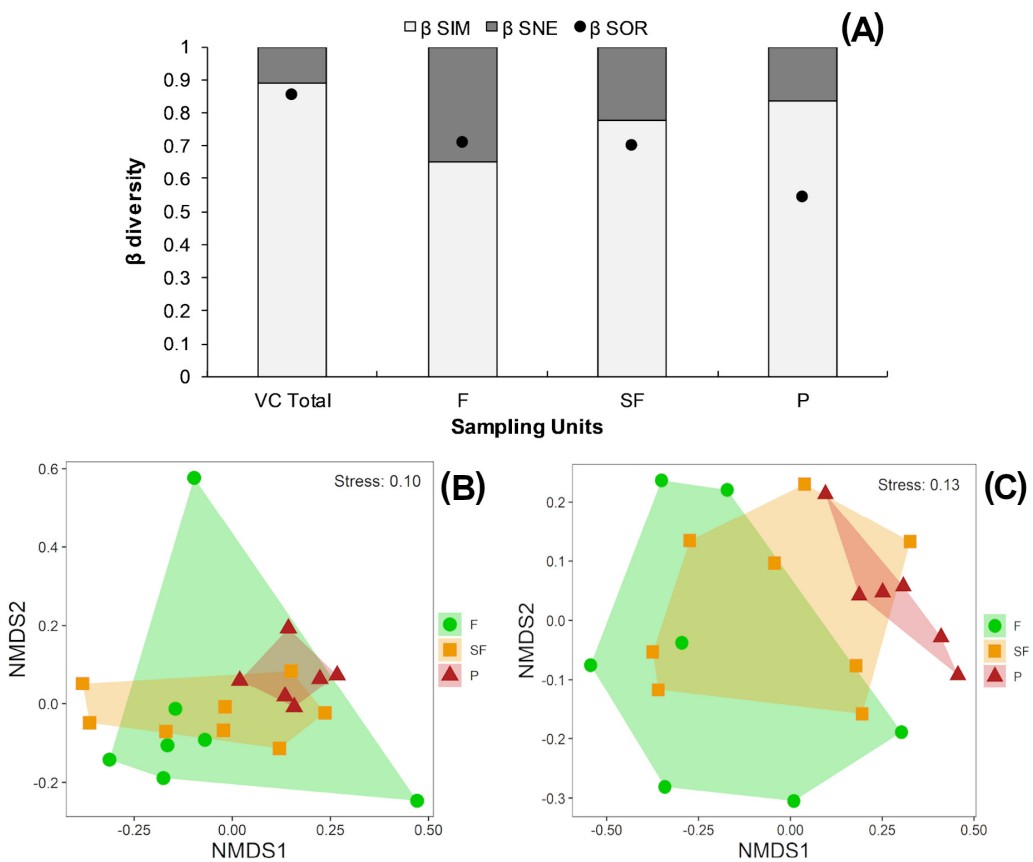

**Figure 4** **(A) Beta diversity within vegetation classes; Non-metric Multidimensional scaling ordination for vegetation classes based on (B) Sørensen dissimilarity & (C) βR-C dissimilarity.** (A) Black dots: Sørensen beta diversity ($\beta$Sor) between vegetation classes. White bars: Percentage contribution of species turnover ($\beta$Sim) to beta diversity ($\beta$Sim/$\beta$Sor), black bars: Percentage contribution of species-nestedness ($\beta$Sne) to beta diversity ($\beta$Sne/$\beta$Sor); (B & C) F, Forest; SF, Second-growth forest; P, Pasture.

variegated windows in the REBISO can be attributed to the convergence of multiple dung beetle assemblages including forest specialists (e.g., *Eurysternus caribaeus, Sulcophanaeus chryseicollis,* and *Uroxys boneti*), forest-pasture edge specialists (*O. landolti, Canthon cyanellus*), open habitat specialists (*D. annae, C. lugubris, O. corrosus*), and generalist beetles (*O. batesi*) (*Favila, 2005*; *Navarrete & Halffter, 2008*).

Intact and fragmented windows showed lower diversity values than variegated windows. This diversity pattern is consistent with the intermediate disturbance theory (*Grime, 1973*). Sites with little or no disturbance favor the predominance of highly competitive forest specialists such as *D. mexicanum, C. vazquezae,* and *E. maya,* which accounted for 85% of the total abundance and 90% of the total biomass in intact windows, thus preventing a higher local diversity. On the other hand, the intense landscape changes caused by livestock production in fragmented windows reduce the local species richness of dung beetles since many native-forest species are unable to adapt to the new open habitat conditions (*Halffter, Favila & Halffter, 1992*; *Silva, Storck-Tonon & Vaz-de Mello, 2016*; *Alvarado et al., 2018*).

The presence of the exotic African species *Digitonthophagus gazella* (*Montes de Oca & Halffter, 1998*) in the REBISO is worth mentioning. Although *D. gazella* was only recorded in pastures of fragmented windows (W6, W7), and contributed with only a small fraction of the community abundance and biomass (six and four percent, respectively), they may pose competitive pressure on native species inhabiting open areas (*Lobo & Montes de Oca, 1994*). Further studies are needed to assess how this invasive beetle might affect native species in the REBISO.

Dung beetles are involved, among other ecological processes, in the recycling of organic matter, soil bioturbation, and secondary seed dispersal (*Nichols et al., 2008*). The amounts of soil removed, dung buried, and seed dispersed are significantly and positively influenced by the species richness and biomass of dung beetle assemblages (*Nunes et al., 2018*; *Alvarado, Dáttilo & Escobar, 2019*). The tropical forest sites showed the highest dung beetle species richness and biomass values. Besides, forest coverage was positively related to dung beetle biomass in the REBISO. Both results indicate that the tropical forest sites likely contain the most functionally efficient dung beetle assemblages, thus emphasizing the importance of forest conservation in the REBISO.

## Effects of landscape composition and configuration on dung beetle assemblages

Previous studies conducted in tropical ecosystems have identified landscape composition as the main predictor of the diversity of dung beetle assemblages (*Sánchez-de Jesús et al., 2016*; *Alvarado et al., 2018*). However, in our study, landscape fragmentation was the primary explanatory variable of variations in species richness and diversity. These findings are likely due to the variegated structure of the REBISO landscape and its "biodiversity-friendly" matrix of second-growth forest (see *Perfecto & Vandermeer, 2008*; *Melo et al., 2013*). First, second-growth forests in the REBISO are structurally similar to forest habitats (*Ramírez-Marcial et al., 2017*). Therefore, while many dung beetle species are restricted to forest patches, others may persist and use the second-growth forest matrix to move between forest patches (*Díaz, Galante & Favila, 2010*). Second, fragmentation in variegated environments creates conditions that allow the coexistence of species from different habitat types (e.g., forest species, pasture species, edge specialist species), thereby increasing the diversity of dung beetles at the landscape scale (*Villada-Bedoya et al., 2016*; *Fahrig, 2017*).

Not all fragmentation effects are beneficial since a higher edge density can have adverse effects on the abundance, biomass, and even the physiological condition of tropical dung beetles (*Portela Salomão et al., 2018*). We found the lowest abundance of dung beetles in variegated windows, where the highest edge density occurs. A higher edge density is coupled with less habitat area, limiting the capacity of the landscape to support medium and large-sized mammal species. *Pozo-Montuy et al. (2019)* observed that medium- and large-sized mammals are significantly less abundant and diverse in the REBISO buffer zone (i.e., where the variegated windows are located). Such reduction in mammal density can cause a marked decrease in dung quantity and availability, thus limiting the growth of dung beetle populations (*Nichols et al., 2009*). Also, microclimatic conditions such as temperature and relative humidity are more variable in forest edges, which might

negatively affect the reproduction and survival of dung beetles (*Klein, 1989*; *Feer, 2013*). Our findings suggest that fragmentation processes in variegated windows foster a high dung beetle diversity, but might also limit their population growth due to insufficient resources, reduced habitat area, or sub-optimal microclimatic conditions. Future studies should assess the strength and extent of this trade-off between dung beetle diversity and abundance, and its functional consequences across the REBISO.

## Beta diversity patterns and mechanisms of diversity maintenance

Species turnover is the primary driver of the high diversity and complementarity of the dung beetle communities found in the REBISO. There were between 3 and 27 species not shared between windows, and from 4 to 35 species not shared between vegetation classes. Each window and vegetation class contributed two or three unique species to the overall diversity. The largest turnover values were found between the fragmented windows (W6, W7, and W8) vs. the variegated and intact windows (W1 to W5), and between the forested vegetation classes (tropical forest, second-growth forest) vs. the pasture sites. The anthropic disturbances and the heterogeneous landscape of REBISO favor this high beta diversity since dung beetles are especially susceptible to environmental variability (*Arellano, Leon-Cortes & Halffter, 2008*; *Costa et al., 2017*).

The differences observed in the species assemblages of fragmented windows (W6, W7, and W8) and those in the other windows (W1 to W5) are not random. Likewise, the differences between tropical forest and pasture assemblages are not random. Significant deviations from random expectations of beta diversity indicate niche-structured assemblages in which environmental filters determine species membership in a community (*Chase et al., 2011*; *Püttker et al., 2015*). Intensive anthropic disturbances such as deforestation can act as an environmental filter in fragmented windows and pastures, selecting stress-tolerant dung beetle species able to survive in open habitats (*Halffter, Favila & Halffter, 1992*; *Spector & Ayzama, 2003*; *Gardner et al., 2008*). *Deltochilum mexicanum*, *C. vazquezae*, *S. chryseicollis*, *Canthon femoralis*, and *E. maya* probably are the species most sensitive to the environmental filters caused by anthropic disturbance. Although these forest species are widely distributed in the biosphere reserve (*Sánchez-Hernández et al., 2018*), their abundance was drastically reduced in fragmented windows.

We found signs of biotic homogenization in the pasture sites. For instance, the lowest alfa and beta diversity values were recorded in pastures, and their species assemblages were more similar to each other than expected by chance, regardless of the windows where they were located, indicating shared environmental filtering processes (*Chase, 2010*). Anthropogenic environmental filters are one of the main drivers of biotic homogenization, eroding alfa and beta diversity and diminishing ecosystem resilience and viability (*Gámez-Virués et al., 2015*). Hence, the advance of the agricultural frontier in the REBISO landscape should be monitored closely to prevent further biotic homogenization processes among the dung beetle species assemblages.

In our study, $^1\beta$ between the intact and variegated windows (W1 to W5), as well as between tropical forests and second-growth forests, was higher than $^0\beta$. Thus, the true beta diversity is mainly due to differences in the abundance of shared species rather

than to differences in richness (*Jost, 2007*). Besides, the overall beta diversity between these windows and vegetation classes was not different from that expected by chance. Most species in neutral communities are considered ecologically equivalent since, in the absence of any factor limiting their dispersal, they can appear at random in any of the null assemblages (*Püttker et al., 2015*; *Ortega-Martínez et al., 2020*). Both results suggest that the REBISO still holds sufficient vegetation cover to maintain a continuous flow of dung beetles between forested landscape sections (W1 to W5).

Given the significant stochasticity of beta diversity between intact and variegated windows, and between tropical forests and second-growth forests, we can conclude that the landscape variegation in the REBISO does not affect dung beetle diversity negatively. However, it is essential to conserve the forested patches to maintain a high dispersal between sites, thereby increasing the resilience of dung beetle populations to habitat loss and isolation (*De Castro Solar et al., 2015*; *Socolar et al., 2016*). Landscape variegation can be an effective strategy to buffer the impact of intense anthropic disturbances (*Rös, Escobar & Halffter, 2012*; *Costa et al., 2017*). Variegation can be achieved by maintaining the forest cover and incorporating more biodiversity-friendly production systems, such as agroforestry practices, in the landscape matrix (*Perfecto & Vandermeer, 2008*).

## CONCLUSIONS

This research contributes to better understand how diversity is distributed in variegated landscapes, and the role of heterogeneous landscapes in the conservation and management of tropical biodiversity. Tropical forests and second-growth forests contributed significantly to maintaining the diversity and biomass of dung beetle assemblages. The variegated structure of the landscape fosters a high dung beetle diversity. The heterogeneity of the REBISO landscape favors the formation of complementary dung beetle communities. Both deterministic and stochastic processes drive the beta diversity patterns in the landscape. Intense anthropic disturbances in fragmented windows and pastures act as non-stochastic filters upon dung beetle species, eroding the alpha and beta diversity of these sites. By contrast, random processes govern the less disturbed sites of the REBISO: fragmented tropical forests and second-growth forests. Increasing habitat variegation in highly disturbed sites can be an effective strategy to buffer and prevent further biotic homogenization processes.

## ACKNOWLEDGEMENTS

We thank María Guadalupe Hernández López and Erick Hernández Baltazar for their assistance with data collection in the field. We thank Paula Enríquez, Alfonso González, Guillermo Ibarra, and three anonymous reviewers for their valuable comments and suggestions. We thank the communities of Emilio Rabasa, Nuevo San Juan Chamula, San Joaquin, and Tierra Nueva for graciously granting access to their land to carry out our fieldwork. We also thank REBISO/CONANP for providing logistic support. María Elena Sánchez-Salazar edited the English version of the manuscript.

### Funding

This research was funded by CONACYT grant PDCPN2013-01-214654 - "Biological and social vulnerability to climate change in Selva El Ocote Biosphere Reserve". The postgraduate studies of José Daniel Rivera at Colegio de la Frontera Sur, San Cristóbal de las Casas, Chiapas, Mexico were supported by scholarships from CONACYT and the Heinrich Böll Stiftung Foundation. The funders had no role in study design, data collection and analysis, decision to publish, or preparation of the manuscript.

### Grant Disclosures

The following grant information was disclosed by the authors:
CONACYT: PDCPN2013-01-214654.
Biological and social vulnerability to climate change in Selva El Ocote Biosphere Reserve.
Heinrich Böll Stiftung Foundation.

### Competing Interests

The authors declare there are no competing interests.

### Author Contributions

- Jose D. Rivera conceived and designed the experiments, performed the experiments, analyzed the data, prepared figures and/or tables, authored or reviewed drafts of the paper, and approved the final draft.
- Benigno Gómez and Lorena Ruíz-Montoya conceived and designed the experiments, authored or reviewed drafts of the paper, and approved the final draft.
- Darío A. Navarrete-Gutiérrez and Mario E. Favila conceived and designed the experiments, analyzed the data, prepared figures and/or tables, authored or reviewed drafts of the paper, and approved the final draft.
- Leonardo Delgado analyzed the data, authored or reviewed drafts of the paper, and approved the final draft.

### Field Study Permissions

The following information was supplied relating to field study approvals (i.e., approving body and any reference numbers):

The Secretaria de Medio Ambiente y Recursos Naturales, México approved field sampling (sgpa/dgs/14214/15).

### Data Availability

Data are available as Supplemental Material.

### Supplemental Information

Supplemental information for this article can be found online at http://dx.doi.org/10.7717/peerj.9860#supplemental-information.

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
