# Peer review of "Mechanisms of diversity maintenance in dung beetle assemblages in a heterogeneous tropical landscape"

_PeerJ, doi:10.7717/peerj.9860_

## Round 0.1 · original submission · Major Revisions

All three reviewers agree that your research is of high values and, after some thorough re-assessment and re-writing - should be in a publishable form for PeerJ. Please review all the comments thoroughly - all three reviewers have put substantive efforts into reviewing your paper, so please make good use of their expertise.

Reviewer 1 ·

Basic reporting

Basic report

1. English: English is good and the technical/scientific terms are employed properly. It is adequate for a wide audience.
2. Intro & Background: the introduction is well structured and brings a relevant background about the main problem addressed in this study.
3. Structure: the study is well structured, with primary and secondary sections presented properly.
4. Figures: some figures could be improved. Below, I present a detailed description of my suggestions (see Minor issues).
5. Raw data: I appreciate the authors provided raw data. However, supporting information has no titles and it is difficult to read apart.

Experimental design

Experimental design

1. Original primary research: this type of research is not totally new but needed because there are not many studies evaluating dung beetles under these approaches.
2. Research question: the central question is relevant; also, the specific questions are clear and adequate.
3. Technical & Ethical standards: Samplings were carried out using standard methods to collect dung beetles and permitted by legal institutions. Besides, voucher material was deposited in a public institution.
4. Methods: in general, the methods are well presented. However, I would like to see a more detailed presentation of the samplings at each site. For instance, the authors have sites, within vegetation classes, within landscape windows. The analytical approach takes into account vegetation classes and landscape windows but they are not totally independent because some vegetation classes only occur in more fragmented landscapes. How do the authors evaluate this kind of dependence?

Validity of the findings

Validity of findings

1. Novelty: the results are good and novel. But I suggest the authors considering a better presentation of both approaches used. Some results are confusing, especially those related to species turnover.
2. Data: the raw data were provided. They are robust and statistically sound.
3. Speculation: the authors based their line of argumentation on their results.
4. Conclusions: they are ok, but a reinterpretation of results based on a better presentation of both analytical approaches used to evaluate the central questions of this study is needed.

Additional comments

Assessment of the manuscript entitled “Mechanisms of diversity maintenance of dung beetle assemblages in a heterogeneous tropical landscape” for PeerJ (peerj-46967)

Main comments

In this paper, the authors aimed to evaluate how the diversity (species richness and Shannon diversity), abundance, and biomass of dung beetles vary across the heterogeneous tropical landscape of Selva El Ocote Biosphere Reserve, Mexico. This study has a good sample design and data to investigate the effects of fragmentation on dung beetles. However, I think that the analytical issues and interpretation of the results should be separated by the two approaches used: spatial approach (windows) and structural approach (classes of vegetation within the windows), as they are not as independent since certain classes of vegetation will be more represented or will occur only in certain classes of fragmentation (e.g. pasture). I raise this issue because the species turnover is said to be random or non-random, depending on the approach used, and this result causes some confusion in the reader. How can the same ecological process be random or non-random? Authors need to improve the presentation and discussion of both approaches so that the reader can better understand the results. I would also like to have a more in-depth discussion of the two approaches and the conservation implications of each. After addressing these issues, the paper will be a good contribution to fragmentation ecology. As follows I present a detailed evaluation.

Abstract
1. Line 25-26. I am not sure about vegetation classes. Wouldn't “secondary vegetation” be a type of “tropical forest”? The difference is the successional stage only.
2. Line 27-28. Perhaps providing average and standard deviation (or error) values could be a better option here because the values sometimes overlap each other when the authors talk about higher and lower species richness: “The highest species richness was recorded in variegated windows (28-37 species) and tropical forest (45 species), while the lowest values were found in intact windows (22-24 species) and pasture (34 species).”
3. Line 31. Why would it be a “random turnover”? Do authors think that intact forests would not have species restricted to this type of habitat?
4. Line 32. I also disagree with the following statement as an explanation to the high beta diversity: “this suggests a high level of connectivity in the landscape.” If there is a high beta diversity between landscape components or habitats, the connectivity among them, measured by compositional changes of dung beetles, is not high if beta diversity dominates. If turnover dominates beta diversity it means some species are restricted spatially, which consequently decreases connectivity.

Introduction
1. Line 66. I suggest the authors be more specific about what “secondary vegetation” does mean. It could be a secondary tropical forest, right? Or is it a different thing?

Methods
1. Line 97. The authors can remove spaces between degrees. There are some coordinates with and without spaces between degrees.
2. Line 104. Again, what does “secondary vegetation” truly mean? Is it a secondary forest? Please, be clear about this issue.
3. Line 113. Add a comma after ‘et al.’
4. Line 126. See also da Silva and Hernández (2015, Plos One, https://doi.org/10.1371/journal.pone.0126112) about trapping distance.
5. Line 146-154. Why do authors use two different methods for the same purpose? Anne Chao’s approach is a more robust approach than using Estimates, which is no longer being updated (see http://viceroy.eeb.uconn.edu/estimates/).
6. Line 159. Change “allow” to “account”
7. Line 163. Ok, but do you use species richness and/or Shannon diversity? It is not totally clear.
8. Line 164-166. Do you mean “betapart R package”? Besides, the nestedness accounted for this metric does not take into account richness differences per se. Therefore, if you have strong richness differences (lack of species shared) and not nestedness, the turnover component estimated by Baselga’s approach is overestimated. See Legendre (2014, Global Ecology and Biogeography, https://doi.org/10.1111/geb.12207) for options, if needed.

Results
1. Line 190. You see, here the authors use “secondary forest” not “secondary vegetation” as a vegetation class.
2. Line 199. Add a space here “E.maya”
3. Line 205. “mexicanus” is italicized.
4. Line 214. Add a space here “df=”
5. Line 216. Add a space here “df=”
6. Line 239. Add a space here “C.vazquezae”
7. Line 243. Add a space here “C.corythus”
8. Line 247. Add a space here “df=”
9. Line 250. Add a space here “df=” and here “P=”
10. Line 251. “secondary vegetation” or “secondary forest”?
11. Line 258. Correct to “nestedness processes”
12. Line 259. “secondary vegetation” or “secondary forest”?

Discussion
1. Line 270. Why do authors think there is a “stochastic distribution of species” here?
2. Line 280. Well, the intermediate disturbance theory is not Connell's. See Wilkinson’s work about “The Disturbing History of Intermediate Disturbance” (Oikos, 84(1): 145-147; https://www.jstor.org/stable/3546874?seq=1).
3. Line 292. Add a space here “E.maya”
4. Line 296. Correct “Digitonthotophagus” to “Digitonthophagus”
5. Line 310. Using both “Beta (β)” is redundant.
6. Line 313. My comment about how nestedness was calculated has its foundation here. Is there at least one species shred between windows and habitats? If not, the turnover would be overestimated since richness differences are not accounted for.
7. Line 321-322. Well, earlier the authors stated that species distributions are random or stochastic! E.g. line 31-32: “random turnover of species between intact and variegated windows, and between tropical forests and secondary vegetation”
8. Line 327. “Deltochilum” is italicized.
9. Line 339. Well, the same diversity process cannot be random and non-random at the same time! There is something going on with the analysis or interpretation of results. The authors need to clarify this issue properly.
10. Line 343-344. Same as the previous one!
11. Line 249-352. I got confused! Previously, you stated “Species turnover between fragmented (W6, W7, and W8) and other windows (W1 to W5) is not a random process; likewise, the differences in diversity between tropical forest and pasture sites are not random.” (Lines 320-322).

Figure 2. I would remove the external line of fig 2a. W7’s symbol has a different line color and size. Besides, why is the fig. 2a limited to ~9000 if you have a total of 15,457 specimens? Shouldn't it be the pooled abundance of all sampled windows? What does the vertical line in fig. 2b mean?

Figure 3. Fig. 3a is not good to read. I suggest separating it into two: one for abundance, one for biomass. Fig. 3n – title: “between windows” or within windows?

Figure 4. Fig. 4a is not good to read. They overlap each other. I suggest separating it into two: one for abundance, one for biomass.

Table 1. Tropical forest description: “2017” is not italicized. Secondary vegetation description: “2017” is not italicized. Pasture description: “± s.e.” with point and after the number. “%” symbol with no spaces after the number.

Table 2. Remove “alpha, beta and gamma” from “Overall alpha, beta and gamma diversity over all the windows” and “Overall alpha, beta and gamma diversity over all the vegetation classes”; it is implicit in the table. Do not forget to specify what “F”, “SV” and “P” mean.

Table 3. Do not forget to specify what “F”, “SV” and “P” mean.

Reviewer 2 ·

Basic reporting

Some moderate revision of the written English is necessary. There are a few issues with verb tenses or run on sentences throughout the manuscript that affects the clarity of the text.
They do have good references, but other references might be added to contrast or support their discussion.
The structure, figures and tables are ok, just minimal editing will be needed.
Very relevant results, but the general discussion lacks of substance, they contrast their results with other publications, but more information and discussion about their results will be better to improve the manuscript.

Experimental design

The authors follow the common and used methodology for this kind of studies and group. Some minimal details that need to be explained in detail are suggested.
This research is done in a Biosphere Reserve, and their results should be used to improve the conservation efforts. They have a lot of interesting findings but not addressed them in a circular discussion or conclusion.
Methodology is good, some clarification is suggested in specific parts of the text.

Validity of the findings

Interesting results for a very well study group.
Authors should take advantage of the natural history and biology of the group, that some of the authors have, to describe and discuss with more detail their results.
Discussion and conclusion can be improved.

Additional comments

The authors studied dung beetles’ diversity patterns and their maintenance mechanisms in a Biosphere reserve at the south of Mexico. Their study has a good methodological approach, from the fieldwork protocol through the diversity analyses. Their results are very interesting and can be better exploited. A broader discussion will be better, there is a lot of potential in them and you can go further with your findings. Especially when talking about the higher diversity in variegated windows in a Biosphere Reserve, they can take advantage of their information, and discuss more conscientiously their results in terms of conservation efforts. Overall the manuscript is very good and with valuable information.

Below are some remarks, comments and questions that I believe must be addressed prior to consider publication to improve the manuscript.

-First, some moderate revision of the written English is necessary. There are a few issues with verb tenses or run on sentences throughout the manuscript that affects the clarity of the text.
-Be consistent with the use of the entire genus name along the text, you change the way you use it, even in the same paragraph.
-Review again the biology and preferences of some of the species, there are some inconsistencies between your discussion and what other authors found.

-Did you try to make your analysis between seasons or differentiating the traps? There were species strictly necrophagous? It might be good to take the functionality; there are two different functional guilds and can react differentially to heterogeneity.

-D. gazella was just collected in W7 and W8, just in pastures, what does that implies for the other sites and for an introduced species like this one. This species might compete with native species inhabited open areas. Read: Lobo, J. M., & Montes, D. O. (1994). Local distribution and coexistence of Digitonthophagus gazella (Fabricius, 1787) and Onthophagus batesi Howden & Cartwright, 1963 (Coleoptera: Scarabaeidae). Elytron, 8, 117-127.

Line 78. read and include this reference here and in some other parts of the text: Alvarado, F., Salomão, R. P., Hernandez-Rivera, Á., & de Araujo Lira, A. F. (2020). Different responses of dung beetle diversity and feeding guilds from natural and disturbed habitats across a subtropical elevational gradient. Acta Oecologica, 104, 103533.

Line 79. In the introduction, you talk about how this landscape analyses can contribute to conservation and diversity maintenance, but later in the discussion, you do not detail this and do not give any suggestions. If you will maintain this phrase in the introduction (use some references too), a larger discussion is expected.

Line 116. since the title you refer to dung beetles, but you use carrion traps too, why? Scarabaeidae that feed on carrion did not fall in the pitfall traps baited with dung? If you sample and analyse carrion beetles too, maybe mention that since the title?

Line 128. carrion traps were also in place 24 hours, did you leave the squid fermenting before setting the traps? Explain more, usually carrion traps are for longer periods.

Line 129. you collect a lot of individuals, you might want to deposit some of the material in other national collections, or to the different coauthors institutions

Line 137. Repetitive phrase, rewrite it.

Line 140. The cite is already in the phrase, no need to cite it again at the end.

Line 150. Why Ros et al cite is there?

Line 198. Why do you think that happen? Maybe the sampling was not enough? How would that affect your results interpretation?

Line 315. Are there any endemic or rare species?

Line 326. Explain what might happens the other way around, with the species from open spaces that cannot go into the forests.

Line 330. Do you mean in REBISO, in Chiapas or in the country? Can you give more details of why this happen? Sampling again or another reason?

Line 338. The difference in abundance seems to be the main trigger for beta diversity changes, but did you analyse different seasons or traps separately? Abundance can be very tricky when using different sampling methods or with seasonality.

Line 344. Could that mean that some species move from one vegetation to the other one, and just some of them will stay only in the forest?

Line 345. Which are the typical species for open spaces or pastures? Are they native or endemic from the state or introduced? Could they benefit from the agricultural/livestock areas, or from other landscape changes?

Line 346. It seem that C. leechi and O. landolti are not typical from open sites but rather forest indicators? (see Alvarado et al 2020). If you collected them in open areas, what would that mean?

Line 348. Species might be just rare in this vegetation, or they are typical in other place. Rephrase avoiding the use of “still minimal”.

Fig1. It is not clear the trap location in each window, it might not be necessary to include that on the map, but is marked in the symbology and is not obvious in the figure. Colors of the vegetation classes are not quite the same in the symbology that in the map, homogenize that. In the rest of the text you refer to the Windows with W and a number, use the same labels for the figure to make it easier to identify them instead of a,b,c,d…

Fig 2. It is not clear what the * means aside the numbers (species identity). I do not have the Figure legends, and this information should be clearly explain there, as well as the meaning of every abbreviation and letters used in the figures.

The table legends are not in the text, some explanation and detail is needed in order to understand them correctly and easily. Details in the supplementary materials are there, I recommend making it too for the other tables.

Table 1 In the vegetation class there are some Windows that repeat, do you mean you separate the traps of every vegetation class to do the analyses? I assume that but is not very clear near in the text, nor the table.

Tables S4a and S4b All species names should be in italics. Verify the authors of each species and homogenise information in both tables, there are some mistakes (e.g. D. amplicollis, P. perplexus, S. chryseicollis); Why there are missing values or abbreviations? ; Abbreviations should be with points and not commas (e.g. Onthophagus maya); Why NA?

Extra literature to read and include in your work and discussion:
Amézquita, S., & Favila, M. E. (2011). Carrion removal rates and diel activity of necrophagous beetles (Coleoptera: Scarabaeinae) in a fragmented tropical rain forest. Environmental entomology, 40(2), 239-246.
Arellano, L., Leon-Cortes, J. L., & Halffter, G. (2008). Response of dung beetle assemblages to landscape structure in remnant natural and modified habitats in southern Mexico. Insect Conservation and Diversity, 1(4), 253-262.
Moctezuma, V., Halffter, G., & Arriaga-Jiménez, A. (2018). Archipelago reserves, a new option to protect montane entomofauna and beta-diverse ecosystems. Revista Mexicana de Biodiversidad, 89(3), 927-937.
Morón, M. A. (1987). The necrophagous Scarabaeinae beetles (Coleoptera: Scarabaeidae) from a coffee plantation in Chiapas, Mexico: habits and phenology. The Coleopterists' Bulletin, 225-232.
Silva, R. J., Storck-Tonon, D., & Vaz-de-Mello, F. Z. (2016). Dung beetle (Coleoptera: Scarabaeinae) persistence in Amazonian forest fragments and adjacent pastures: biogeographic implications for alpha and beta diversity. Journal of insect conservation, 20(4), 549-564.

Reviewer 3 ·

Basic reporting

Overall this paper was well written, with sufficient references. However, I feel that a bit more context and clarity are needed to make the paper suitable for publication.

The abstract needs work. A sentence or two of background at the start would help to set the stage before jumping into details of results. See the following link for some guidelines.
https://www.wiley.com/network/researchers/preparing-your-article/how-to-write-a-scientific-abstract

I also feel that the Introduction needs to include a bit more background. I think it would be good to include more context about habitat vs landscape - your paper is about looking at biodiversity at the landscape level, but most biodiversity research is looked at from a habitat level so elaborate more on how your study is different. Can you clarify your third research question i.e. how is it different from question 2?

Could you clarify the windows concept and how it links with landscapes? Windows are an important part of the project yet they are not mentioned in the introduction. In the introduction, you talk about landscapes and vegetation but there is no mention of windows. Not all of your readers will have a solid background in all the ecological terms so it would be helpful to add a bit more clarity and background.

The results are well presented, but Figure 1 needs some work. Why do a and b have smaller circles than the others. Figure design is confusing - could you use black instead of white for letters OR perhaps use lines to connect to the location instead of letters? Why not highlight the circle with black so its is easier to see. I think more explanation is needed in the figure legend. You could explain what category each window represents. Is window centre important? From the map legend: trap location is not used so should not be included in legend. Can you create more contrast between the orange and the yellow boxes (secondary and pasture) to match better with map image?
Thank you for supplemental data tables, however, each table need a title with a number. While legends are optional, I think a brief explanation of what the table is showing would be very helpful.
You also mention communities (line 253) but these are not explained anywhere.

You mentioned that dung beetles were sampled in both rainy season and dry season, but didn't mention how this may have affected the results. You sampled using two different baits but didn't mention how this may have affected the results either.

In the discussion, species turnover is mentioned as an important biodiversity measure, but it’s not mentioned in introduction or methods. Could this be explained further?

Experimental design

I feel that the research is relevant and meaningful. Dung beetles make excellent bioindicators and can be very useful for increasing our knowledge for conservation management. The research questions were well defined. The sampling efforts were commendable and combined with extensive statistical analysis. Methods were described sufficiently with the exception of how biomass was determined.

Validity of the findings

The data appear to be sound. Null models were discussed.

The conclusions are stated clearly but don’t quite match with research questions. However, this is simply a matter of clarity, not a problem.

Additional comments

Overall this is a good paper. A lot of work went into sampling and statistical analysis. The text just needs a bit more clarity to effectively communicate the results and implications of this research.

Annotated reviews are not available for download in order to protect the identity of reviewers who chose to remain anonymous.

---

## Round 0.2 · Minor Revisions

Great to see the substantive improvements to the manuscript. There are still some minor issues to deal with before we accept the manuscript for publication.

Reviewer 1 ·

Basic reporting

no comment

Experimental design

no comment

Validity of the findings

no comment

Additional comments

After my second reading of the paper, I was able to notice the great improvement of the study in theoretical and analytical terms. The authors addressed my major and minor suggestions. As in my previous assessment, I reaffirm the quality of the study in terms of all the necessary aspects to be assessed (basic report, experimental design, and validity of the findings). However, a few issues of writing still remain. After these corrections, I think the paper will be a good contribution to the understanding of fragmentation ecology in variegated landscapes.

Minor issues
Line 40. Change “beta-diversity” to “beta diversity”
Line 219. Change “beta-diversity” to “beta diversity”
Line 252. Lack of space “E.maya”
Line 283. Change “nested processes” to “nestedness-resultant component”
Line 349. Lack of space “O.batesi”
Line 352. Remove a space here “is consistent”
Line 471. Change “beta-diversity” to “beta diversity”

---

## Round 0.3 · accepted · Accept

Congrats Jose and team for a great manuscript. All the changes you have made to the manuscript since submission have substantially strengthened it, and I wish you well in promoting your work, and enabling this research to be widely read and cited!.
best wishes
Nigel